# Ampicillin-susceptible *Enterococcus faecium* infections: clinical features, causal clades, and contribution of MALDI-TOF to early detection

Hélène Faury,[1,2] Ronan Le Guen,[1,2] Vanessa Demontant,[3] Christophe Rodriguez,[3] Bérénice Souhail,[4] Adrien Galy,[4] Sarah Jolivet,[5] Raphaël Lepeule,[4] Jean-Winoc Decousser,[2,6] Vincent Fihman,[1] Paul-Louis Woerther,[1,3,6] Guilhem Royer[1,6,7]

**ABSTRACT**   *Enterococcus faecium,* a common resident of the human gastrointestinal tract, is also a major pathogen. Prompt initiation of appropriate treatment is essential to improve patient outcome in disseminated *E. faecium* infections. However, ampicillin resistance is frequent in this species, rendering treatment difficult. We used a comprehensive approach, including clinical data review, whole-genome sequencing, and mass spectrometry, to characterize ampicillin-susceptible (EFM-S) and ampicillin-resistant (EFM-R) isolates. We included all patients with culture-confirmed *E. faecium* infection attending our hospital over a 16-month period. A comparison of 32 patients infected with EFM-S strains and 251 patients infected with EFM-R strains revealed that EFM-R isolates were strongly associated with a longer hospital stay, history of prior hospitalization, and the carriage of multidrug-resistant organisms. An analysis of the genomes of 26 EFM-S and 26 EFM-R isolates from paired patients revealed a population structure almost perfectly matching ampicillin susceptibility, with resistant isolates in clade A1, and susceptible isolates in clades A2 and B. The clade B and A2 isolates mostly came from digestive or biliary tract samples, whereas clade A1 isolates were mostly obtained from urine and blood. Finally, we built a custom database for matrix-assisted laser desorption/ionization time-of-flight mass spectrometry (MALDI-TOF MS), which differentiated between clade B and clade A1/A2 strains with high-positive and high-negative predictive values (95.6% and 100%, respectively). This study provides important new insight into the clinical features and clades associated with EFM-S and EFM-R isolates. In combination with MALDI-TOF MS, these data could facilitate the rapid initiation of the most appropriate treatment.

**IMPORTANCE** *Enterococcus faecium* is an important human pathogen in which the prevalence of ampicillin resistance is high. However, little is known about the clinical characteristics of patients infected with ampicillin-resistant and ampicillin-susceptible strains. Indeed, current knowledge is based on genus-wide studies of *Enterococcus* or studies of very small numbers of susceptible isolates, precluding robust conclusions. Our data highlight specific clinical features related to the epidemiology of EFM-S and EFM-R strains, such as length of hospital stay, history of prior hospitalization, carriage of multidrug-resistant organisms, and type of sample from which the isolate was obtained. The use of matrix-assisted laser desorption/ionization time-of-flight mass spectrometry with a custom-built database may make it possible to distinguish clade B isolates, which are typically susceptible to ampicillin, from clade A1/A2 isolates (A1 being typically resistant), thereby facilitating the management of these infections.

**KEYWORDS**   *Enterococcus faecium*, MALDI-TOF, whole-genome sequencing, ampicillin susceptibility

Address correspondence to Guilhem Royer, guilhem.royer@aphp.fr.

Hélène Faury and Ronan Le Guen are joint first authors. Author order was determined alphabetically.

The authors declare no conflict of interest.

See the funding table on p. 11.

Enterococci are Gram-positive cocci belonging to phylum Firmicutes, which includes more than 60 different species. Two of these species, *Enterococcus faecalis* and *Enterococcus faecium,* are the most prevalent species in humans. They are closely associated with the human intestinal microbiota and may cause a wide range of infections (1). *E. faecium* has been divided into two main clades: A and B (2–4). Clade A contains hospital- and animal-associated isolates, which were initially classified into two different clades: A1 and A2, respectively (2). Given the particular adaptation capacities of *E. faecium*, it was suggested that these specific clones emerged from animal strains under antibiotic-selective pressure about 75 years ago (2, 5). However, further studies on larger numbers of strains did not support this subdivision, instead identifying a basal group of strains corresponding to clade A2 and a rapidly evolving clone corresponding to clade A1 (3). By contrast, clade B is mainly associated with community-acquired infections (6). Some authors have even recently proposed that it should be reassigned to species *Enterococcus lactis* (7).

The antimicrobial arsenal for dealing with enterococci is limited. As a result, empirical treatment options for enterococcal infections are also generally limited and based on broad-spectrum antibiotics active against Gram-positive bacteria, such as vancomycin. Indeed, vancomycin is frequently required to treat multidrug-resistant strains, such as ampicillin-resistant *E. faecium* clones from hospitals (8). In this species, the overproduction of low-affinity penicillin-binding protein 5 (PBP5) is frequently associated with a decrease in susceptibility to ampicillin. This phenomenon, which is dependent on both rearrangements upstream from *pbp5* and point mutations/insertions (mostly at positions 466 and 485), is closely associated with the ampicillin-resistant strains of the A1 clade, and sometimes of the A2 clade, and is not observed in the ampicillin-susceptible strains of clade B (9). *E. faecium* bacteremia increased between 2001 and 2014 in Europe and between 2001 and 2010 in the United States (10). This increase may have led to an increase in the use of broad-spectrum antibiotics (e.g., vancomycin) as an empirical option in hospitals, in the absence of rapid results concerning antibiotic susceptibility. The excessive use of broad-spectrum antibiotics not only has ecological consequences but may also not be the most effective therapeutic approach. Conflicting results have been published (11), but some studies have reported better outcomes with amoxicillin-based treatments in patients infected with susceptible strains (12, 13).

In this context, it would be useful to be able to precisely identify the *E. faecium* strains responsible for severe infections. The clinical and epidemiological data associated with resistance (e.g., nosocomial acquisition) (14) must be taken into account when selecting the most appropriate treatment. However, tools accelerating the identification of enterococcal species could also provide a robust and complementary means of improving this choice.

The objective of this study was to identify the risk factors associated with the isolation of ampicillin-susceptible (EFM-S) or -resistant (EFM-R) *E. faecium* from infected patients and to determine whether susceptibility could be predicted from the precise identification of *E. faecium* to clade level by MALDI-TOF mass spectrometry as part of the routine bacterial identification protocol.

## MATERIALS AND METHODS

### Study design

We retrospectively included all patients with a documented *E. faecium* infection tested for ampicillin susceptibility between 1 January 2020 and 20 April 2021 at the Bacteriology Unit of Henri Mondor Hospital (Créteil, France), a 2,814-bed tertiary-care hospital. Isolates obtained during screening for vancomycin-resistant *E. faecium* (VRE) carriage were excluded. During the study period, bacterial identification was routinely performed with MALDI-TOF MS (matrix-assisted laser desorption/ionization time-of-flight mass spectrometry), with the MALDI Biotyper system (Bruker). Ampicillin susceptibility was determined by the disk diffusion method, with a disk loaded with 2 µg of ampicillin, or

by the broth microdilution method (MicroScanSystem, Beckman Coulter). The isolates were classified as susceptible or resistant to ampicillin according to CA-SFM (*Comité de l'Antibiogramme de la Société Française de Microbiologie*, CA-SFM V2.0 Mai 2019) recommendations based on EUCAST (European Committee on Antimicrobial Susceptibility Testing) guidelines.

We first focused on the clinical characteristics of the patients infected with EFM-S or EFM-R isolates. We took into account only the first episode for each patient included. We reviewed the demographic and clinical data for the patients with electronic medical records and a standardized form (Table S1). The data collected included sex, age, comorbid conditions (cardiovascular risk factors, solid cancer, hemopathy, cardiovascular diseases, autoimmune and systemic diseases, solid organ transplantation, and chronic kidney disease), prior hospitalization for more than 24 hours over the preceding year, reasons for consultation, multidrug-resistant organism (MDRO) carriage. We also collected data about the length of stay in the hospital and the medical ward in which the patient was staying at the time of sampling. Clinical outcome (i.e., vital status) was monitored for up to 30 days, for all the included patients.

We then compared the sources of isolates and the polymicrobial nature of the infection as a function of ampicillin susceptibility. In this context, we included all isolates from all episodes occurring in all patients over the study period. Only redundant isolates obtained from the same clinical sample during a given episode were excluded. We encountered no cases of differences in susceptibility between isolates collected from the same source during the same episode.

## Collection of isolates for whole-genome and MALDI-TOF MS analysis

All available EFM-S isolates ($n = 26$) identified over the survey period were thawed and paired with EFM-R isolates ($n = 26$) from patients of the same sex and similar age. The identification obtained with the MALDI-TOF MS Biotyper system (Bruker) and ampicillin susceptibility (disk diffusion method) was checked for all these isolates by the broth microdilution method (see above). The isolates were then subjected to whole-genome sequencing on an Illumina NextSeq sequencer (paired-end 150 bp reads).

## Whole-genome analysis

Genomes were assembled with shovill 1.0.4 (15), SPAdes v3.13.1 (16), trimmomatic v0.39 (17), lighter v1.1.2 (18), FLASH v1.2.11 (19), bwa v0.7.17 (20), pilon v1.23 (21), and samtools v1.9 (22). The quality of the assemblies was then checked with CheckM (23) using the taxonomy-specific workflow (species "*E. faecium*") to ensure a completeness exceeding 95% and less than 5% contamination. Multi-locus sequence typing (MLST) was performed with mlst (24). We also retrieved the 72 genomes representative of the diversity within *E. faecium* available from the study by Lebreton et al. (2) to obtain a broader overview of this species. These genome sequences were combined with our newly sequenced genomes, and Roary v3.12.0 (25) with standard parameters was used to generate a pangenome. The core gene alignment was used to construct a phylogenetic tree with FastTree v2.1.8 (26), with a general time-reversible evolution model and a gamma distribution of rates across sites. This tree was annotated with Itol (27) and was used to assign the genomes to the previously described clades A1, A2, and B. We also searched for mutations of PBP5, focusing on the two positions close to the active site considered to be responsible for increasing resistance to ampicillin (i.e., insertion after S466 and/or mutation at position M485) (6, 28–31). We also characterized the genetic environment of *pbp5* based on the nomenclature proposed by Montealegre et al. (9). A new letter was assigned to each new environment detected. In a few genomes, the *pbp5* gene or its upstream and downstream sequences were too fragmented to define the environment (Table S2). Whole-genome sequences are available from Bioproject PRJEB56579.

## Construction and evaluation of the custom-built MALDI-TOF MS database

Based on the susceptibility testing results and the whole-genome sequencing analysis, 10 EFM-R and 17 EFM-S isolates corresponding to clades A1 ($n = 10$), A2 ($n = 10$), and B ($n = 7$) were selected for the construction of a custom-built MALDI-TOF MS database with MBT Compass RUO v4.1, FlexAnalysis v3.4, MBT compass explorer v4.1, and the MALDI Biotyper (Bruker). Proteins were extracted with the Mass Spectrum Profile (MSP) Creation protocol (V1.1, Bruker). Briefly, we collected fresh colonies with a 1-µL inoculation loop and mixed them with 300 µL LC-MS water. We then denatured the proteins by adding 900 µL pure ethanol. After careful vortexing and centrifugation ($13,000 \times g$, 2 min), we discarded the supernatant and performed another centrifugation to remove the residual ethanol. The pellets were then dried at room temperature and were dissolved and thoroughly mixed in 30 µL each of 70% formic acid and acetonitrile. This solution was then centrifuged at $13,000 \times g$ for 2 min. Finally, we spotted 1 µL of the supernatant onto an MSP 96 target BC steel plate for each of the eight replicates. We allowed the spots to dry and then coated the plate with 1 µL HCCA matrix solution (α-cyano-4-hydroxy-cinnamic acid; Bruker, Germany). Mass spectra were obtained according to the MALDI Biotyper protocol (V.2.4, Bruker) and were carefully inspected with FlexAnalysis software. Spectra with a mass peak deviation >0.05%, outlier peaks, and flatlines were eliminated, and the remaining spectra were combined to generate a single mass spectrum for each strain in MBT compass explorer, with the default parameters. The mass spectra were used to generate a new local database, available from the MALDI Biotyper Compass RUO software. We also constructed a dendrogram from these MSPs on the MALDI Biotyper Compass RUO (Euclidean distance, complete linkage) to check the clustering of isolates.

The performance of the local database was then evaluated according to the routinely used procedure for all 52 isolates, including the 27 isolates used to construct the database. For each strain, colony fragments from overnight cultures on blood agar at $35°C \pm 2°C$ were spotted onto a target and overlaid with 1 µL HCCA-matrix. The spectra obtained on the MALDI biotyper were compared with the data in the newly generated database, with MALDI Biotyper Compass RUO software. The algorithm generated a log score value ranging from 0 to 3.0, with a log score ≥2.0 indicating a "high confidence identification" at species level, according to the MALDI Biotyper Compass IVD protocol (V4.2, Bruker). For an identification to be considered valid, it had to meet the following criteria: a log score ≥2.0 and concordance of the first two identification scores for ampicillin susceptibility and clade membership. Positive and negative predictive values were calculated for each of these parameters separately.

## Statistical analyses

Statistical analyses were performed with GraphPad Prism v.5. and Stata v.16.0. Categorical variables are expressed as numbers (and percentages, %) and were compared in Fisher's exact tests. Continuous variables are expressed as medians (and interquartile range; IQR) and were compared in Mann-Whitney $U$ tests. A $P$-value <0.05 was considered statistically significant.

Logistic regression analysis was performed to identify the factors independently associated with EFM-S isolates. We excluded variables from the multivariate analysis if their $P$-value in the univariate analysis was >0.15. We used the Wald test for systematic evaluations of the interactions between each of the variables in the final model. Collinear variables were excluded from the multivariate analysis. The number of relevant variables reported in the final multivariate model was adapted to the number of events to calculate odds ratios (1 variable: 10 patients).

## RESULTS

### Clinical characteristics of the patients infected with EFM-S and EFM-R

The demographic and clinical characteristics of the 283 patients infected with EFM-S ($n = 32$) or EFM-R ($n = 251$) are shown in Table 1. The proportion of women was higher in

**TABLE 1** Overall baseline characteristics of patients with EFM-S or EFM-R isolates[b]

|  | EFM-S[c] (n = 32) | EFM-R (n = 251) | P-value[d] |
|---|---|---|---|
| Male—no (%) | 13 (40.6) | 157 (62.5) | 0.02 |
| Age—median (IQR), years | 76 (54–83) | 69 (58–80) | 0.33 |
| Hospital department in which sampling occurred—no (%) |  |  |  |
| Surgical unit | 20 (62.5) | 53 (21.1) | <0.0001 |
| Digestive surgery | 16 (50.0) | 19 (7.6) | <0.0001 |
| Urology | 3 (9.4) | 13 (5.2) | 0.40 |
| Vascular surgery | 1 (3.1) | 14 (5.6) | 1.00 |
| Non-surgical unit | 9 (28.1) | 101 (40.2) | 0.25 |
| Hematology | 1 (3.1) | 33 (13.1) | 0.15 |
| Hepato-gastroenterology | 4 (12.5) | 19 (7.6) | 0.31 |
| Oncology | 2 (6.3) | 9 (3.6) | 0.36 |
| Geriatric service | 2 (6.3) | 17 (6.8) | 1.00 |
| Intensive care unit | 1 (3.1) | 60 (23.9) | 0.005 |
| Rehabilitation service | 2 (6.3) | 31 (12.4) | 0.40 |
| Underlying comorbid conditions—no (%) |  |  |  |
| None | 3 (9.4) | 5 (2.0) | 0.05 |
| Cardiovascular risk factors | 16 (50.0) | 168 (66.9) | 0.08 |
| Solid cancer | 11 (34.4) | 67 (26.7) | 0.40 |
| Hemopathy | 2 (6.3) | 58 (23.1) | 0.04 |
| Cardiovascular diseases | 11 (34.4) | 115 (45.8) | 0.26 |
| Autoimmune and systemic disease | 4 (12.5) | 24 (9.6) | 0.54 |
| Solid organ transplant | 1 (3.1) | 21 (8.4) | 0.49 |
| Chronic kidney disease | 5 (15.6) | 32 (12.7) | 0.59 |
| Prior hospitalization within the preceding year—no. (%) | 13 (40.6) | 185/243 (76.1) | <0.0001 |
| MDRO carriage in the preceding year—no. (%) | 1/10 (10.0) | 72/111 (64.9) | 0.001 |
| Duration of hospital stay during which sampling occurred—median (IQR), days | 5 (1–10) | 16 (6–29) | <0.0001 |
| Patients hospitalized for more than 48 hours at the time of sampling—no. (%) | 19 (59.4) | 210 (83.7) | 0.003 |
| Death[a]—no. (%) | 7 (21.9) | 104 (41.4) | 0.04 |

[a]Clinical outcomes (vital status) were monitored until 20 May 2021.
[b]The denominators of the patients included in this analysis are shown if different from the total number for the corresponding group. Each patient was counted only once, and none of the patients belong to both groups.
[c]EFM-S, *E. faecium* ampicillin-susceptible; EFM-R, *E. faecium* ampicillin-resistant; no., number; IQR, interquartile range; MDRO, multidrug-resistant organism.
[d]A P-value <0.05 was considered statistically significant.

the EFM-S group (59.4% vs 37.5%, *P* = 0.02). Patients with EFM-S were more frequently hospitalized in digestive surgery units (50.0% vs 7.6%, *P* < 0.0001). Those with EFM-R were hospitalized for a longer period (16 vs 5 days, *P* < 0.001), were more likely to have a history of prior hospitalization during the preceding year (76.1% vs 40.6%, *P* < 0.0001), and were more frequently MDRO carriers (64.9% vs 10%, *P* = 0.001). Moreover, EFM-R patients were more likely to suffer from hemopathy (23.1% vs 6.3%, *P* = 0.04) and require intensive care (23.9% vs 3.1%, *P* = 0.005).

The multivariate logistic regression analysis revealed that hospitalization in a digestive surgery unit was significantly associated with infection with EFM-S isolates (Table 2). Conversely, prior hospitalization and the duration of hospital stay were associated with the risk of EFM-R detection.

## Characteristics of EFM-S and EFM-R isolates

Overall, 372 isolates of *E. faecium* were grown from various clinical samples during the study period. These isolates included 151 obtained from a patient from whom another isolate was also obtained. There were 61 patients with multiple isolates. We found that 116 of these multiple isolates were associated with the same episode as the other isolate or isolates from the same patient. In total, multiple isolates were available for 48 episodes: 22 episodes involved isolates from different sources (*n* = 48 isolates), 19

**TABLE 2** Factors associated with infection with ampicillin-susceptible EFM isolates

| Variables | Univariate analysis | | Multivariate analysis | |
| --- | --- | --- | --- | --- |
| | OR (95% CI) | *P*-value | OR (95% CI) | *P*-value |
| Male | 0.41 (0.19–0.87) | 0.02 | | |
| Age[a] (years) | 1.00 (0.98–1.03) | 0.65 | | |
| Hospital department at the time of sampling | | | | |
| Digestive surgery | 12.21 (5.29–28.17) | <0.001 | 8.68 (3.36–22.41) | <0.001 |
| ICU[c] | 0.10 (0.01–0.77) | 0.03 | | |
| Underlying comorbid conditions (≥1) | 0.20 (0.04–0.86) | 0.03 | | |
| Hemopathy | 0.22 (0.05–0.96) | 0.04 | | |
| MDRO carriage | 0.06 (0.01–0.51) | 0.01 | | |
| Prior hospitalization during the preceding year | 0.21 (0.10–0.46) | <0.001 | 0.25 (0.10–0.61) | 0.002 |
| Duration of hospital stay during which sampling occurred[b] (days) | 0.92 (0.88–0.96) | <0.001 | 0.92 (0.88–0.97) | 0.002 |

[a]Per one unit increase.
[b]In whole numbers of days for each patient.
[c] ICU, intensive care unit; MDRO, multidrug-resistant organism; OR, odds ration; and CI, confidence interval.

involved isolates from the same source (*n* = 41 isolates), and 7 episodes involved isolates from both the same and different sources (*n* = 27 isolates).

In total, 38 isolates (10.2%) were ampicillin susceptible, and 334 (89.8%) were ampicillin resistant. Most of the EFM-S isolates were isolated from digestive samples (*n* = 27/38; 71.1% vs 15.0% *n* = 50/334 for EFM-R isolates; *P* < 0.0001). EFM-S isolates were more frequently obtained from biliary samples (36.8% vs 4.2%, *P* < 0.0001), peritoneal fluid (15.8% vs 2.7%, *P* = 0.002), and abdominal mass (18.4% vs 5.1%, *P* = 0.006) than EFM-R isolates. They were also more frequently obtained from polymicrobial cultures (78.9% vs 42.2%, *P* < 0.0001). Conversely, EFM-R isolates were more frequently obtained from blood cultures (21.6% vs 5.3%, *P* = 0.02) and urinary samples (50.0% vs 21.1%, *P* = 0.001) than EFM-S isolates. Vancomycin resistance was detected in only two EFM-R isolates and none of the EFM-S isolates.

## Clade organization, genetic variation, and clinical characteristics associated with a subset of 52 isolates

In total, 52 isolates were selected for further analysis, including the 26 isolates available from the 38 EFM-S isolates identified over the survey period. The remaining 12 EFM-S strains were not available due to the short shelf-life of isolates for some samples (e.g., urinary samples). We paired these 26 EFM-S isolates with 26 EFM-R isolates from patients of the same age and sex. All these isolates were then sequenced with Illumina short-read technology and analyzed. Core genome phylogenetic analysis of these genomes and those from the data set of Lebreton et al. (2) identified the three previously described clades: A1, A2, and B (Fig. 1). Two genomes, EnGen0002 and 1_231_408, could not be classified and were considered to be hybrid genomes, as suggested by Lebreton et al. (2). The organization of the population into clades almost perfectly matched the distribution of ampicillin susceptibility of our isolates: 7/7 EFM-S clade B, 26/27 EFM-R in clade A1, and 18/18 EFM-S isolates in clade A2 (Fig. 1; Table S2). As expected, all clade B genomes displayed pattern A for the *pbp5* genetic environment, with no insertion at position 466′ and methionine at position 485 (Fig. S1 and S2; Table S2). Interestingly, the only ampicillin-susceptible genome in clade A1 (EFM-S-25) also displayed pattern A, with no mutations at positions 466′ and 485, ruling out an error in our data set. None of the other genomes in clade A presented pattern A; all were mutated at position 485 (M485A or M485T), and many had an insertion at position 466′ (*n* = 37/48). Finally, the resistant strains of the A2 clade displayed at least two characteristics of the non-A pattern, the insertion at position 466′ and the mutation at position 485. The only exception was the EnGen0052 genome. However, Montealegre et al. had already noted a discrepancy for this strain (9), which they found to display pattern C in a PCR-based approach,

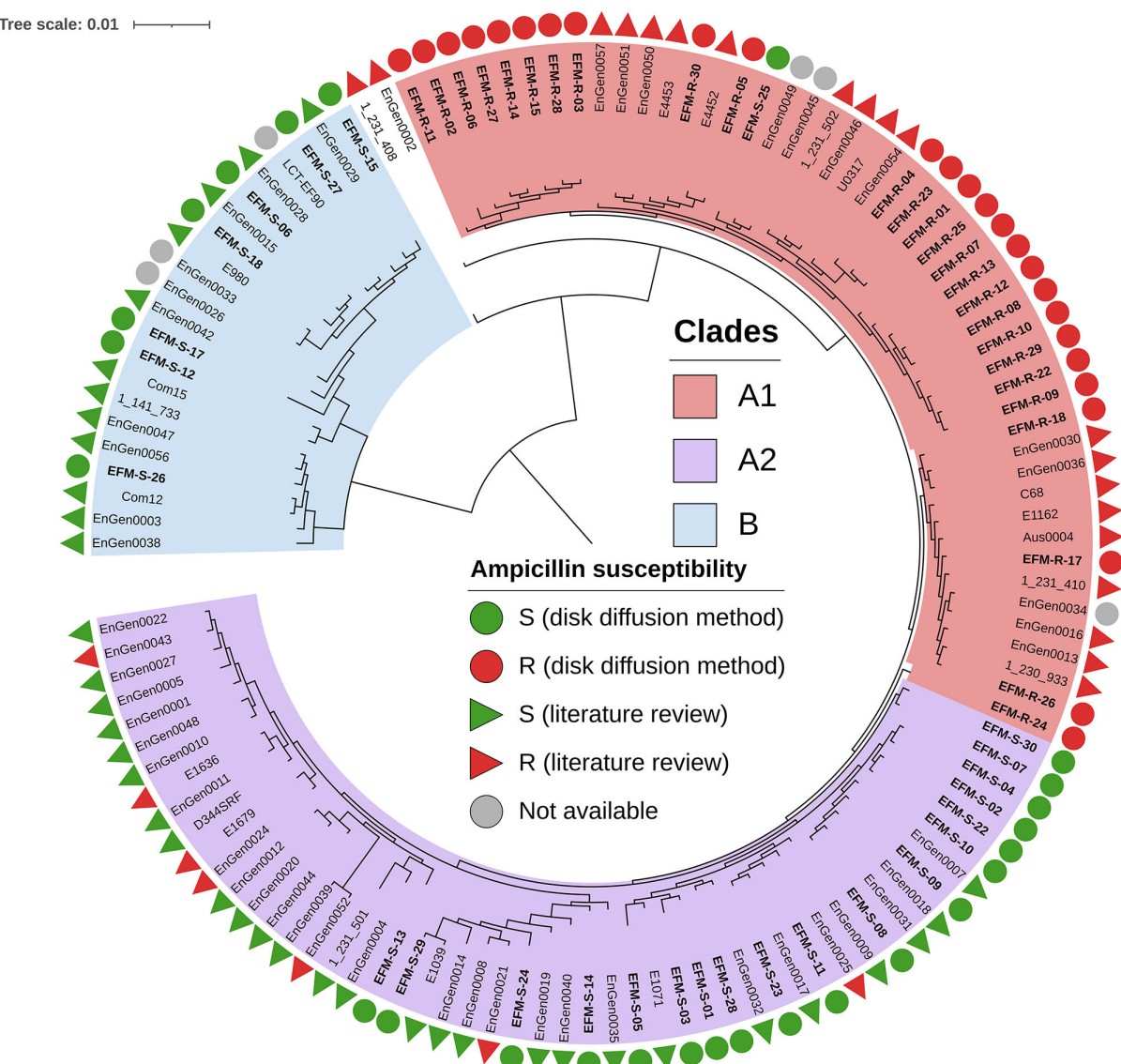

**FIG 1** Core genome-based phylogenetic tree for the isolates. The 52 genomes from our study are shown in bold. The other genomes were obtained from the study by Lebreton et al. (2). The genomes were assigned to one of the previously described clades—A1, A2, or B—highlighted in salmon, purple, and blue, respectively. In the outermost circle, the colored circles and triangles represent ampicillin susceptibility, determined by disk diffusion methods or extracted from previous publications. Gray circles represent missing values. The tree is mid-point rooted between clade B and non-clade B genomes. The scale represents genetic distances in nucleotide substitutions per site.

but pattern A based on whole-genome sequence analysis. We also observed four new patterns that we named F to I. Pattern F was found only in clade A1 ($n = 15$) and corresponds to a shortened *msr* gene (597 bp) with no *ftsW* upstream. Pattern G was also found only in clade A1 ($n = 8$) and consists of a reduced *msr* gene (642 bp) with an upstream IS*Efm1*. Pattern H, found in the A2 ($n = 6$) and A1 ($n = 1$) clades, was similar to pattern B but with no insertion sequence between *msr* and *pbp5*. Finally, pattern I was also similar to pattern B but with an IS*1542* instead of an IS*Efm1*.

Most of the clade A1 isolates were obtained from urine ($n = 15/27$) or blood cultures ($n = 6/27$), and these isolates frequently belonged to ST80 ($n = 8/27$) and ST117 ($n = 11/27$). Conversely, the isolates of clades B and A2 were mostly isolated from digestive or biliary tract-related samples ($n = 6/7$ and $n = 14/18$, respectively) and were more diverse in terms of ST (Table S2).

The clinical characteristics of the patients tended to differ between clades (i.e., for the patients corresponding to the 52 isolates sequenced, Table 3). Patients infected with isolates from clades A2 and B were more frequently hospitalized in digestive surgery units (61.1% and 42.9%, respectively, vs 7.4% in the clade A1) than those infected with clade A1 isolates. Their hospital stay was also shorter (1.5 and 5 days, respectively) than that of patients with clade A1 isolates (22 days). Moreover, prior hospitalization in the preceding year, MDRO carriage, and hospitalization for more than 48 hours were less frequent in the group of patients with clade A2 isolates than in those with clade A1 isolates (33.3% vs 72.0%; 0% vs 70.0%; and 44.4% vs 85.2%, respectively).

## Construction and evaluation of the performance of the MALDI biotyper database

For each of the 27 strains selected for database construction, more than 20 spectra fulfilled the necessary criteria according to the manufacturer's recommendations (i.e., spectra without mass peak deviation >0.05%, outlier peaks, or flatlines). The individual spectra and the database generated from them are publicly available from Zenodo (https://doi.org/10.5281/zenodo.7936571). The dendrogram constructed from the MSPs showed that isolates from clade B clustered on one side and those from clades A1/A2 clustered on the other (Fig. S3). We evaluated our newly created database to distinguish between the 52 isolates in terms of their ampicillin susceptibility and clade (Table 4). Ampicillin susceptibility was correctly predicted for 26/52 isolates (50%). Clade (A1, A2, or B) was correctly identified for 23/52 isolates (44.2%). Finally, in assessments of the ability to differentiate clade B from clades A1/A2, correct predictions were obtained for 50/52 isolates (96.2%).

These good results can be explained by the differences observed between the spectra for clade B and A1/A2 isolates, as shown in Fig. S4. The two discrepancies resulted from (i) EFM-S-06 (clade B) and EFM-S-08 (clade A2) being identified as the best matches in the analysis of EFM-S-06 (clade B) and (ii) EFM-S-06 (clade B) and EFM-S-08 (clade A2) being identified as the best matches in the analysis of EFM-S-29 (clade A2). These discrepancies are thus due to the MSP of the EFM-S-06 isolate, which appears to be different from that of the other isolates of clade B (Fig. S3).

**TABLE 3** Baseline characteristics of patients according to clades (sequenced isolates, $n = 52$)

| | Clade A1 ($n = 27$) | Clade A2 ($n = 18$) | Clade B ($n = 7$) |
|---|---|---|---|
| Male—no.[b] (%) | 12 (44.4) | 8 (44.4) | 4 (57.1) |
| Age—median (IQR), years | 75 (56.5–82.0) | 76.0 (54.0–84.0) | 78.0 (57.0–89.0) |
| Hospital department in which sampling occurred—no. (%) | | | |
| Digestive surgery unit | 2 (7.4) | 11 (61.1) | 3 (42.9) |
| Non-surgical unit | 11 (40.7) | 4 (22.2) | 3 (42.9) |
| Hepato-gastroenterology | 4 (14.8) | 4 (22.2) | 0 |
| Intensive care unit | 4 (14.8) | 1 (5.6) | 0 |
| Rehabilitation service | 5 (18.5) | 1 (5.6) | 0 |
| Underlying comorbid conditions—no. (%) | | | |
| None | 1 (3.7) | 3 (16.7) | 0 |
| Cardiovascular risk factors | 19 (70.3) | 9 (50.0) | 4 (57.1) |
| Solid cancer | 6 (22.2) | 6 (33.3) | 4 (57.1) |
| Hemopathy | 5 (18.5) | 0 | 0 |
| Cardiovascular diseases | 15 (55.5) | 5 (27.8) | 3 (42.9) |
| Prior hospitalization during the preceding year—no. (%) | 18/25 (72.0) | 6 (33.3) | 5 (71.4) |
| MDRO carriage during the preceding year—no. (%) | 7/10 (70.0) | 0/6[a] | 1/3 (33.3) |
| Duration of hospital stay during which sampling occurred—median (IQR), days | 22 (8–46) | 1.5 (0.75–8.5) | 5 (3–15) |
| Patients hospitalized for more than 48 hours at the time of sampling—no. (%) | 23/27 (85.2) | 8/18 (44.4) | 6 (85.7) |

[a]The denominators of patients included in this analysis are shown if they differ from the total number in the corresponding group.
[b]No., number; IQR, interquartile range; MDRO, multidrug-resistant organism.

**TABLE 4** Performance of the local database for distinguishing isolates on the basis of ampicillin susceptibility and clade membership

|  | EFM-R/EFM-S ($n$ = 26/26) | Clade A1 ($n$ = 27) | Clade A2 ($n$ = 18) | Clade B/(A1 + A2) ($n$ = 7/45) |
|---|---|---|---|---|
| Positive predictive value (%) | 26.9 | 25.9 | 52.6 | 95.6 |
| Negative predictive value (%) | 73.1 | 72.0 | 42.4 | 100 |

Based on these results, our ability to distinguish between isolates on the basis of ampicillin susceptibility or clade (A1, A2, and B) appeared very low. However, performances were much better for distinguishing clade B from clade A1/A2, with very high-positive and -negative predictive values (95.6% and 100%, respectively).

## DISCUSSION

We describe here an original analysis of EFM-S and EFM-R isolates responsible for infections based on a combination of clinical data review, whole-genome sequencing, and the characterization of bacterial mass spectra (MALDI-TOF MS). We first show that the patients with infections involving EFM-R can be clearly distinguished from those with infections involving EFM-S by the significantly longer hospital stay, more frequent history of hospitalization within the preceding year, and more frequent MDRO carriage in patients with EFM-R isolates, highlighting the link to the nosocomial environment. The association between ampicillin resistance and the hospital environment has already been established in *Enterococcus* spp. (14, 32), but only in studies considering the entire *Enterococcus* genus and including very few EFM-S isolates, precluding any robust conclusion regarding the particular case of *E. faecium*.

By sequencing and analyzing the genomes of a subset of 52 isolates, consisting of pairs of EFM-R and EFM-S isolates from matched patients, we were also able to integrate the clade to which these isolates belonged and the genetic variation associated with resistance. Indeed, we confirmed that the *E. faecium* population is organized into a limited number of clades—A1, A2, and B—highly congruent with the community or hospital origin of the isolates (2, 5, 33). Our data set also confirmed the association of clade B with ampicillin susceptibility and of clade A1 with ampicillin resistance. Clade A2 has been shown to contain both susceptible and resistant isolates (9). The observation exclusively of susceptible isolates in this clade in our data set may reflect the limited number of isolates studied and/or the particular epidemiological features of our hospital. The observed genetic variants were well correlated with phenotype, with combinations of at least two of three features (non-A pattern, insertion after position 466 and mutation at position M485) being found in all resistant genomes. The inferred clades of our isolates also revealed epidemiological trends in the clinical characteristics of the patients from whom they were obtained. Nosocomial infection was suggested for patients with infections involving clade A1 isolates, who had a longer hospital stay, and were more likely to have been hospitalized in the preceding year and to display MDRO carriage. Interestingly, isolates from clades B and A2 were more frequently encountered in the samples of digestive or biliary tract origin, consistent with the commensal nature of the isolates of clade B (2). Such differences were also recently observed by Arredondo-Alonso et al. who highlighted the low prevalence of non-A1 isolates in hospitalized patients (4). However, these data do not reflect a link between resistance and virulence but rather the potential adaptation of clades to a specific environment.

Finally, we constructed a custom database for routine identification with the MALDI Biotyper (Bruker) system, using a subset of 27 isolates, and we evaluated its performance on 52 isolates. The proximity of the sister clades A1 and A2 made it impossible to discriminate between them. Consequently, we were also unable to distinguish between isolates on the basis of ampicillin susceptibility profile at the whole-population scale. Nevertheless, this database performed well for discriminating clade B isolates from those of clades A1/A2. These results were expected, given the significant genetic divergence of clade B from clades A1/A2, suggesting that they could almost be considered to belong to two different species (2, 7). In terms of clinical microbiology, this implies that the

detection of EFM-S isolates by MALDI-TOF MS can be achieved only by identifying the subpopulation corresponding to clade B. Further investigations are required to identify peaks potentially specific for ampicillin resistance, as has been proposed for VRE detection (34, 35). However, it may be difficult to follow such an approach with the same protocol as for bacterial identification (HCCA matrix, m/z range 2,000–20,000) due to the mechanism of ampicillin resistance (overexpression and/or decreased affinity).

Clade B isolates appear to be less frequent than clade A1/A2 isolates in human infections, but their rapid and accurate identification in routine laboratory experiments may make it possible to prescribe the most appropriate antibiotic treatment (i.e., ampicillin) straight away. Indeed, some studies have suggested that the use of vancomycin rather than ampicillin may lead to a poorer outcome in ampicillin-sensitive *Enterococcus* infections (12, 13). Further studies are required to address this issue in the particular case of EFM-S and EFM-R strains. The use of MALDI-TOF MS to distinguish between clades B and A1/A2 could facilitate the setting up of prospective studies in this context.

Our study has several limitations, including its retrospective, single-center nature. Larger prospective studies, including more diverse hospitals and patients, will be required to confirm our results. Future studies should also try to decipher and confirm the clinical characteristics of patients associated with each clade through robust statistical tests. In addition, the inclusion of a larger number of isolates would make it possible to improve the MALDI-TOF database, to prevent discrepancies due to a divergent isolate from clade B. Nevertheless, the results obtained are consistent with the known ampicillin susceptibility profiles and clades of the isolates. Moreover, to the best of our knowledge, this is one of the few studies focusing solely on the *E. faecium* species. The lower frequency of EFM-S (10.2% here) isolates reported here may account for the lack of published data.

In conclusion, we think that combining clinical data with bacterial identification via our custom-built MALDI-TOF MS database would make it easier to distinguish between EFM-S and EFM-R isolates. MALDI-TOF MS is widely available for bacterial identification in clinical bacteriology laboratories. Our database could, therefore, be incorporated into routine use, facilitating early initiation of the most appropriate treatment. The early use of a narrow-spectrum antibiotic, such as ampicillin instead of glycopeptides, would prevent exposure to broad-spectrum antibiotics even for as little as 24 or 48 hours, which is known to increase the risk of acquiring multidrug-resistant bacteria.

## ACKNOWLEDGMENTS

We thank Céline Sakr and Florence Cizeau from the Equipe Opérationnelle d'Hygiène (Département de prévention, diagnostic et traitement des infections, AP-HP, Hôpital Henri Mondor, Créteil, France) for technical assistance.

G.R. was partially supported by the Agence Nationale de la Recherche under the French National Action Plan on Antimicrobial Resistance (PAMR)—project SEQ2DIAG (ANR-20-PAMR-0010).

We declare that we have no conflicts of interest.

## AUTHOR AFFILIATIONS

[1]Unité de Bactériologie, Département de Prévention, Diagnostic et Traitement des Infections, AP-HP, Hôpital Henri Mondor, Créteil, France

[2]Equipe Opérationnelle d'Hygiène, Département de Prévention, Diagnostic et Traitement des Infections, AP-HP, Hôpital Henri Mondor, Créteil, France

[3]Plateforme de Séquençage Haut-Débit, Département de Prévention, Diagnostic et Traitement des Infections, AP-HP, Hôpital Henri Mondor, Créteil, France

[4]Unité Transversale de Traitement des Infections (U2TI), Département de Prévention, Diagnostic et Traitement des Infections, AP-HP, Hôpital Henri Mondor, Créteil, France

[5]Unité de Prévention du Risque Infectieux, AP-HP, Hôpital Saint-Antoine, Paris, France

[6]EA 7380, Université Paris-Est Créteil, Ecole Nationale Vétérinaire d'Alfort, USC Anses, Créteil, France

[7]EERA Unit "Ecology and Evolution of Antibiotic Resistance", Institut Pasteur-Assistance Publique/Hôpitaux de Paris-Université Paris-Saclay, Paris, France

## AUTHOR ORCIDs

Jean-Winoc Decousser ⓘ http://orcid.org/0000-0001-7105-6202
Guilhem Royer ⓘ http://orcid.org/0000-0001-9092-8336

## FUNDING

| Funder | Grant(s) | Author(s) |
|---|---|---|
| Agence Nationale de la Recherche (ANR) | ANR-20-PAMR-0010 | Guilhem Royer |

## AUTHOR CONTRIBUTIONS

Hélène Faury, Formal analysis, Investigation, Methodology, Writing – original draft | Ronan Le Guen, Formal analysis, Investigation, Methodology, Writing – original draft | Vanessa Demontant, Formal analysis, Resources | Christophe Rodriguez, Resources, Supervision | Bérénice Souhail, Formal analysis, Resources | Adrien Galy, Formal analysis, Resources | Sarah Jolivet, Formal analysis, Methodology, Supervision | Raphaël Lepeule, Formal analysis, Resources | Jean-Winoc Decousser, Resources, Supervision | Vincent Fihman, Resources, Supervision | Paul-Louis Woerther, Conceptualization, Methodology, project administration, Writing – review and editing | Guilhem Royer, Conceptualization, Data curation, Investigation, Methodology, Supervision, Visualization, Writing – review and editing

## DATA AVAILABILITY

The genomes sequenced as part of this study are available from Bioproject PRJEB56579. The individual spectra and the MALDI-TOF database generated as part of this study are publicly available from Zenodo.

## ETHICS APPROVAL

This study was approved by the institutional review board of Henri Mondor University Hospital (2023-151).

## ADDITIONAL FILES

The following material is available online.

### Supplemental Material

**Figure S1 (Spectrum04545-22-S0001.pdf).** Genetic environment of pbp5 and distribution of isolates according to clades and ampicillin susceptibility.
**Figure S2 (Spectrum04545-22-s0002.pdf).** Coregenome-based phylogenetic tree of the 52 isolates from our study and the 72 isolates from the study by Lebreton et al.
**Figure S3 (Spectrum04545-22-s0003.pdf).** Dendrogram calculated from the Mass Spectra Profiles (MSP) of the 27 isolates used to construct the MALDI-TOF MS database.
**Figure S4 (Spectrum04545-22-s0004.pdf).** Example of mass spectra ranges obtained from the 27 isolates used to build the MALDI-TOF MS database
**Table S1 (Spectrum04545-22-s0005.xlsx).** Demographic and clinical data collected for each patient.
**Table S2 (Spectrum04545-22-s0006.xlsx).** Characteristics of the isolates.

## Open Peer Review

**PEER REVIEW HISTORY (Reviewer history.pdf).** An accounting of the reviewer comments and feedback.

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
