## [Reviewer comments · Microbiology Spectrum]

Microbiology Spectrum

Ampicillin-susceptible *Enterococcus faecium* infections: clinical features, causal clades and contribution of MALDI-TOF to early detection

Hélène Faury, Ronan Leguen, Vanessa Demontant, Christophe Rodriguez, Bérénice Souhail, Adrien Galy, Sarah Jolivet, Raphael Lepeule, Jean-Winoc Decousser, Vincent Fihman, Paul-Louis Woerther, and Guilhem Royer

Corresponding Author(s): Guilhem Royer, APHP - Hôpital Henri Mondor - Département de Microbiologie

Review Timeline:

Submission Date:	November 8, 2022
Editorial Decision:	February 6, 2023
Revision Received:	June 6, 2023
Accepted:	July 28, 2023

Editor: Vittal Ponraj

Reviewer(s): Disclosure of reviewer identity is with reference to reviewer comments included in decision letter(s). The following individuals involved in review of your submission have agreed to reveal their identity: Dhammika H Navarathna (Reviewer #3)

Transaction Report:

DOI: <https://doi.org/10.1128/spectrum.04545-22>

February 6, 2023

Dr. Guilhem Royer
APHP - Hôpital Henri Mondor - Département de Microbiologie
Bactériologie
51 avenue du Maréchal de Lattre de Tassigny
Créteil 94010
France

Re: Spectrum04545-22 (Ampicillin susceptible *Enterococcus faecium* infections: clinical features, causative clades and contribution of MALDI-TOF to early detection)

Dear Dr. Guilhem Royer:

Thank you for submitting your manuscript to Microbiology Spectrum. We are willing to consider a revised version of this paper at Spectrum, it would be in your best interest to improve the writing. You are also welcome to use one of the services here: <https://journals.asm.org/content/language-editing-services>. You may consider using these services when revising your manuscript. The use of these services will have no direct bearing on the editorial decision. ASM has no affiliation with these companies.

Link Not Available

Sincerely,

Vittal Prakash Ph.D S(ASCP)

Journals Department
Reviewer comments:

Reviewer #1 (Comments for the Author):

The authors studied *Enterococcus faecium* strains for their resistance to ampicillin using genome analysis and MALDI-TOF MS.

Major comments

1. The susceptibility to antimicrobial agents in *Enterococcus* does not predict virulence or clinical significance. *E. faecalis* that is mostly susceptible to ampicillin is a more virulent species, and causes more infective endocarditis, compared to *E. faecium* that is mostly ampicillin resistant. While the clade map data were understandable, the conclusion on categorizing specimen source in this study was over-generalized or over-simplified. For examples, in Suppl. Table 1, of the 16 strains of *E. faecium* isolated from blood that were reported in the literature, 6 were in clade A1, 6 were in clade A2, and the remaining 4 were in clade B. This result contrasted the finding in this report and the statement in lines 269 -271.
2. The association between the clade and the susceptibility to ampicillin was still weak. Of the 52 strains in clade A2 (figure 1), 13.5% (7 of 52) showed very major error (false susceptibility), 2 of which were collected from blood.
3. It may be more direct to identify ampicillin resistance using MALDI-TOF MS, similar to the approach to identify VRE, than correlating MALDI-TOF spectra to the clade distribution to guide treatment.

Minor comments

1. Lines 64 and 74: Please clarify "last-line antibiotic"
2. Lines 75-78: The sentence was unclear
3. Line 87: MALDI-TOF MS instead of electronic mass spectrometry
4. Line 98: Bruker software version and manufacturer information should be included

Reviewer #2 (Comments for the Author):

Faury et al. describe the clinical features, causative clades, and the use of the MALDI-TOF method to detect *E. faecium* at one hospital in France. The study is notable due to the lack of data to date in this area, with implications for clinical management of identified infections. Some comments, questions and suggestions for improvement are outlined below:

1. The absence of mutation/genetic variant data for the strains sequenced in this study is a missed opportunity. Given that the authors performed WGS, this data could be invaluable to confirm and evaluate resistance profiles within and between clades and potentially identify key variants for targeted molecular drug susceptibility testing. This information would strengthen the study and also add to the suggestions for clinical management of infections.
2. It is unclear why the authors chose to focus solely on ampicillin susceptibility, including screening for resistance only to this drug, as the inclusion of additional drug information (incl. to glycopeptides, linezolid and daptomycin, etc) might again help to define clades as well as to guide clinical management of infections.
3. Why weren't 2 results grouped correctly via MALDI? Some further discussion of this point is necessary.
4. The limitations of this work are notable, including, but not limited to, the inclusion of strains at only 1 hospital (from 1 setting) over 15mo, the focus on a single drug, and the inability to compare features between the identified clades with any statistical power. All should be discussed.
5. Some comment on how feasible the suggested algorithm actually is in most hospital settings would be informative

Minor comments:

- Explain Asterix in table 3
- The A2 (susceptible) grouping doesn't match the ref in the intro and should be mentioned
- Please provide ref for low frequency of clade B
- Please review text of submission for English and grammatical errors

Reviewer #3 (Comments for the Author):

Hélène Faury et al., pursued a timely investigation of Ampicillin susceptible/resistant *Enterococcus faecium* infections in clinical setting addressing causative clades and exploring MALDI-TOF to help early prediction of susceptibility or resistance.

Unfortunately, inability to discriminate clade A1 vs A2 from MALDI-TOF shattered the applicability. Now it is best to give more weight for WGS. In addition their samples WGS data should be able to access from public data base.

Change the wording of the topic as you compared both sensitivity and resistance for the outcome of the study.

Introduction is poorly written, and I believe this is basically gathering data from original publications and converting to your own words to avoid copy right issues. When authors write their own idea independently, the flow is very nice, e.g., Line 84-88.

Please read the literature and get the big pic and write in your own words and then cite what you wrote.

Line 64: list the last line antibiotics to the benefit of the student readers and residents. Rewrite the complete sentence from line 63-65.

Line 68- "rearrangement" should be grammatically corrected as "rearranged".

Line 67: before you mention about Clades, introduce *E faecalis* clade system in a separate sentence.

Line 73-75: not clear. rewrite the sentence starting "today".

Line 76: inadequate treatment does not make any sense. Did you imply treatment failure?

Line 80: required sounds mandatory. Replace with "helpful". Rewrite the sentence starting with "In addition" as it does not pass the idea well.

Line 105-106: When you do your second part of the analysis, does this mean you used multiple episodes from same patient? Its not acceptable to get multiple isolate per patient within a calendar year for the sensitivity criteria (Read antibiogram construction in CLSI) . In addition, single patient multiple samples can have variable AST over the time. Explain how you overcame this issue.

Line 109: supplemental table .1- you have used only 55 clinical isolates from your study and used previously published strains for the rest. In addition, explain how and when you used 283 patients' data and how many of them are multiple isolates from a single patient?

Line 150: WGS- specify how many strains you sequenced. It's in the results but not in the methods.

Line 144-145: brief MSP creation protocols and add as a supplemental material for the benefit of the readership.

Line 140-153 and 229-243: show common spectra associated with FEM-S and FEM-R. I was unable to open the file in the data base shown in line 233. This could be due to my organization computer restriction. But make sure other can access it. How do you address inability of discriminating isolates between clade A1 and A2 as they are different in Amp sensitivity?

Line 171: What are the criteria used for the P value 0.15? it not a significant number. Could this be a typo?

Line 204-205: What is the basis of selecting 52 isolates? if you are to show the clade specific unique sensitivity, you need every single isolate per patient in your cohort. There is no randomized selection criteria. Any explanation for this? Running more numbers of isolates and library prep is not that difficult these days.

Did you see any other phenotypic difference in clade A1 vs A2? Discuss in the discussion section.

Staff Comments:

Preparing Revision Guidelines

Please return the manuscript within 60 days; if you cannot complete the modification within this time period, please contact me. If you do not wish to modify the manuscript and prefer to submit it to another journal, please notify me of your decision immediately so that the manuscript may be formally withdrawn from consideration by Microbiology Spectrum.

Reviewer comments:

Reviewer #1 (Comments for the Author):

The authors studied *Enterococcus faecium* strains for their resistance to ampicillin using genome analysis and MALDI-TOF MS.

Major comments

1. The susceptibility to antimicrobial agents in *Enterococcus* does not predict virulence or clinical significance. *E. faecalis* that is mostly susceptible to ampicillin is a more virulent species, and causes more infective endocarditis, compared to *E. faecium* that is mostly ampicillin resistant. While the clade map data were understandable, the conclusion on categorizing specimen source in this study was over-generalized or over-simplified. For examples, in Suppl. Table 1, of the 16 strains of *E. faecium* isolated from blood that were reported in the literature, 6 were in clade A1, 6 were in clade A2, and the remaining 4 were in clade B. This result contrasted the finding in this report and the statement in lines 269 - 271.

We totally agree with the reviewer that resistance should not be interpreted as a proxy of virulence. However, we did not draw such a parallel in our study. Nonetheless, we observed that ampicillin susceptible isolates were more frequently found in digestive samples whereas EFM-R isolates were more frequently found in blood culture. In our dataset, it is associated to the enrichment in clade B and clade A2, which are more frequently susceptible to ampicillin than clade A1 and are known as commensal. Indeed, over the 26 out of the 38 EFM-S that we were able to sequence, all belonged to clade B (n=7) or A2 (n=18).

Concerning the second part of the comment, our isolates were obtained in a well-defined population over a given period of time. On the contrary, the isolates from the literature have been obtained through different studies with various sampling criteria. This is highlighted by Lebreton et al. who gave the following precision about the isolate selection : "Strains selected for genome analysis were drawn from those representing diverse points within the known phylogenic structure, as determined by MLST (Fig. 1), and are listed in Table S1 in the supplemental material". Source information has been added for isolates from the literature in an effort to be comprehensive. But the inclusion of these isolates primarily allows us to examine ampicillin susceptibility across the entire *E. faecium* species and to be consistent with the previously defined clades.

To avoid confusion, we have added the following sentence to the discussion (lines 355-356):

"However, these data do not reflect a link between resistance and virulence, but rather the potential adaptation of clades to a specific environment."

2. The association between the clade and the susceptibility to ampicillin was still weak. Of the 52 strains in clade A2 (figure 1), 13.5% (7 of 52) showed very major error (false susceptibility), 2 of which were collected from blood.

Indeed, we agree with the reviewer. This is the reason why we conclude that only the rapid identification of clade B can be considered as a proxy for ampicillin susceptibility (lines 365-

367): "In terms of clinical microbiology, this implies that the detection of EFM-S isolates by MALDI-TOF MS can be achieved only by identifying the subpopulation corresponding to clade B".

Furthermore, it is known from the literature that the A2 clade contains both susceptible and resistant isolated (PMID: 27821450), as already mentioned in the discussion (lines 341-342).

3. It may be more direct to identify ampicillin resistance using MALDI-TOF MS, similar to the approach to identify VRE, than correlating MALDI-TOF spectra to the clade distribution to guide treatment.

We agree that direct identification of ampicillin susceptibility using MALDI-TOF would be a very interesting approach. However, ampicillin resistance is due to a decreased affinity and/or an increased expression of *pbp5* whereas vancomycin resistance is associated with the acquisition of new genes (*vanA/vanB*) leading to new proteins directly identifiable through mass spectra analysis. Therefore, it is improbable that MALDI-TOF could help in detecting ampicillin susceptibility.

Furthermore, recent studies have indeed shown that it is possible to directly differentiate VRE from VSE using the Maldi TOF (PMID: 35204419, PMID 34756065), but the discriminatory power remains low for clinical application (e.g. recall of 0.79 in the study by Wang et al.) compared for example to PCR-based approach.

To make this point more clear, we added the following comment on the need for a direct approach (lines 367-371):

"Further investigations are required to identify peaks potentially specific for ampicillin resistance, as has been proposed for VRE detection (34, 35). However, it may be difficult to follow such an approach with the same protocol as for bacterial identification (HCCA matrix, m/z range 2,000-20,000) due to the mechanism of ampicillin resistance (overexpression and/or decreased affinity)."

Minor comments 1. Lines 64 and 74: Please clarify "last-line antibiotic"

We now change the sentences to "broad-spectrum antibiotic against Gram-positive bacteria, such as vancomycin" and "broad-spectrum antibiotic (e.g. vancomycin)".

2. Lines 75-78: The sentence was unclear

We have modified the sentence as follows (lines 86-90):

"The excessive use of broad-spectrum antibiotics not only has ecological consequences but may also not be the most effective therapeutic approach. Conflicting results have been published (11), but some studies have reported better outcomes with amoxicillin-based treatments in patients infected with susceptible strains (12, 13)."

3. Line 87: MALDI-TOF MS instead of electronic mass spectrometry

Done

4. Line 98: Bruker software version and manufacturer information should be included

We now added the version of the softwares used (line 170).

Reviewer #2 (Comments for the Author):

Faury et al. describe the clinical features, causative clades, and the use of the MALDI-TOF method to detect *E. faecium* at one hospital in France. The study is notable due to the lack of data to date in this area, with implications for clinical management of identified infections. Some comments, questions and suggestions for improvement are outlined below:

1. The absence of mutation/genetic variant data for the strains sequenced in this study is a missed opportunity. Given that the authors performed WGS, this data could be invaluable to confirm and evaluate resistance profiles within and between clades and potentially identify key variants for targeted molecular drug susceptibility testing. This information would strengthen the study and also add to the suggestions for clinical management of infections.

We thank you the reviewer for this suggestion. We initially did not search for genetic variants associated with ampicillin resistance as it has already been broadly studied by several authors. Indeed, few mutations/genetic variations within or upstream *pbp5* have been shown to be repeatedly linked to ampicillin resistance in the literature.

As requested by the reviewer, we search for these genetic variations among the genomes: insertion at position 466', mutation at position 485 and genetic environment of *pbp5* [according to the nomenclature proposed by Montealegre et al. (cf Figure 2 from PMID: 27821450)]. Of note we found 4 new patterns. All the patterns found in our dataset and the distribution of clade A1/A2/B and ampicillin S/R are shown in figure S1. We also provide a phylogenetic tree as figure S2. This tree gathered all information about clades, susceptibility, genetic environment and mutations at position 466' and 485.

As expected, all clade B genomes showed pattern A for the *pbp5* genetic environment, no insertion at position 466' and a methionine at position 485. Interestingly, the only ampicillin-susceptible genome in clade A1 (EFM-S-25) also shows pattern A and has no mutations at positions 466' and 485, ruling out any errors in our dataset. The other A1 clade genomes never showed the A pattern, are all mutated at position 485 (M485A or M485T) and frequently have an insertion at position 466' (n=37/48). Finally, the resistant strains of the A2 clade showed at least two characteristics among the non-A pattern, the insertion at position 466' and the mutation at position 485. The only exception was the EnGen0052 genome. However, Montealegre et al. had already noted a discrepancy for this strain: they found the pattern C by a PCR-based approach, whereas it appears to carry the pattern A based on whole genome sequence analysis.

These new analysis have been added in the material and methods, results and discussion sections.

2. It is unclear why the authors chose to focus solely on ampicillin susceptibility, including screening for resistance only to this drug, as the inclusion of additional drug information (incl. to glycopeptides, linezolid and daptomycin, etc) might again help to define clades as well as to guide clinical management of infections.

We focused only on ampicillin susceptibility because it is a narrow-spectrum antibiotic that is usually used as first-line treatment of susceptible isolate responsible for invasive and non-invasive infections.

We agree with the reviewer that further information on other drugs could also be very useful. However, when looking back at the isolate data, we found very few isolates resistant to vancomycin (n=2/372), linezolid (n=2/372) or daptomycin (n=11 over the 282 isolates tested). In this context, the very limited number of observations prevents any firm conclusion on these resistant isolates and was out of the scope of this study.

Moreover, the prevalence of VRE is still very low in France (0.8%) according to the ECDC (<https://atlas.ecdc.europa.eu/public/index.aspx?Dataset=27&HealthTopic=4>).

3. Why weren't 2 results grouped correctly via MALDI? Some further discussion of this point is necessary.

These two discrepancies were due to the fact that i) EFM-S-06 (clade B) and EFM-S-08 (clade A2) were identified as the best matches when analyzing EFM-S-06 (clade B), and ii) EFM-S-06 (clade B) and EFM-S-08 (clade A2) were identified as the best matches when analyzing EFM-S-29 (clade A2). This is due to differences in the MSP of the EFM-S-06 isolate which appears to be distant from the other isolates of the B (Figure S3).

We have now added these precisions to the results section. We also discuss the need to increase the sampling of strains to improve the efficiency of our database (lines 383-385).

4. The limitations of this work are notable, including, but not limited to, the inclusion of strains at only 1 hospital (from 1 setting) over 15mo, the focus on a single drug, and the inability to compare features between the identified clades with any statistical power. All should be discussed.

We already discussed the retrospective and single-centre nature of our study. We also conclude on the need for bigger prospective studies to help deciphering the clinical characteristics associated with each clade.

We have now added the following sentence to highlight the need for a larger number of isolates to improve our MALDI-TOF database (see point n°3 above) (lines 383-385):

“In addition, the inclusion of a larger number of isolates would make it possible to improve the MALDI-TOF database, to prevent discrepancies due to a divergent isolate from clade B.”

We have also modified the discussion as follow to consider the possibility of bias in our A2 isolates due to the monocentric nature of our study (lines 341-344):

“Clade A2 has been shown to contain both susceptible and resistant isolates (9). The observation exclusively of susceptible isolates in this clade in our dataset may reflect the limited number of isolates studied and/or the particular epidemiological features of our hospital.”

5. Some comment on how feasible the suggested algorithm actually is in most hospital settings would be informative.

We now added the following comment (lines 392-397):

“MALDI-TOF MS is widely available for bacterial identification in clinical bacteriology laboratories. Our database could therefore be incorporated into routine use, facilitating early initiation of the most appropriate treatment. The early use of a narrow-spectrum antibiotic, such as ampicillin instead of glycopeptides, would prevent exposure to broad-spectrum antibiotics even for as little as 24 or 48 hours, which is known to increase the risk of acquiring multidrug-resistant bacteria.”

Minor comments:

-Explain Asterix in table 3

We apologize for the lack of explanation. We have added it to the bottom of the table.

-The A2 (susceptible) grouping doesn't match the ref in the intro and should be mentioned

We now mentioned in the introduction that i) the distinction between A1 and A2 may depend on the size of the dataset, as suggested by Raven et al (PMID: 27527616) and ii) isolates from the clade A2 may sometimes be resistant to ampicillin.

-Please provide ref for low frequency of clade B

We now added the following sentence with reference to the study by Arredondo-Alonso *et al.* (PMID: 32047136) (lines 353-355):

“Such differences were also recently observed by Arredondo-Alonso et al., who highlighted the low prevalence of non-A1 isolates in hospitalized patients (4).”

-Please review text of submission for English and grammatical errors

The manuscript has now been reviewed for English and grammatical errors by expert scientific writers.

Reviewer #3 (Comments for the Author):

Hélène Faury et al., pursued a timely investigation of Ampicillin susceptible/resistant *Enterococcus faecium* infections in clinical setting addressing causative clades and exploring MALDI-TOF to help early prediction of susceptibility or resistance. Unfortunately, inability to discriminate clade A1 vs A2 from MALDI-TOF shattered the applicability. Now it is best to give more weight for WGS. In addition their samples WGS data should be able to access from public data base.

We apologize for this. We have checked and the bioproject PRJEB56579 is now publicly available.

Change the wording of the topic as you compared both sensitivity and resistance for the outcome of the study. Introduction is poorly written, and I believe this is basically gathering data from original publications and converting to your own words to avoid copy right issues. When authors

write their own idea independently, the flow is very nice, e.g., Line 84-88. Please read the literature and get the big pic and write in your own words and then cite what you wrote.

Like many introduction sections, our introduction retrieve data from original publications and tries to extract and gather in our own words the most useful information related with the aims of our study. However, the introduction has been rewritten as requested by the Reviewer.

Line 64: list the last line antibiotics to the benefit of the student readers and residents. Rewrite the complete sentence from line 63-65.

As also requested by reviewer 1, we changed the term "last-line antibiotic" to "broad-spectrum antibiotic against Gram-positive bacteria" and gave the example of vancomycin. We modified the sentence from lines 73-75 as follows:

"The antimicrobial arsenal for dealing with enterococci is limited. As a result, empirical treatment options for enterococcal infections are also generally limited and based on broad-spectrum antibiotics active against Gram-positive bacteria, such as vancomycin."

Line 68- "rearrangement" should be grammatically corrected as "rearranged".

Done.

Line 67: before you mention about Clades, introduce E faecalis clade system in a separate sentence.

We have now introduced the clade system while citing the conflicting views against it (lines 63-72):

"E. faecium has been divided into two main clades: A and B (2–4). Clade A contains hospital- and animal-associated isolates, which were initially classified into two different clades: A1 and A2, respectively (2). Given the particular adaptation capacities of E. faecium, it was suggested that these specific clones emerged from animal strains under antibiotic selective pressure about 75 years ago (2, 5). However, further studies on larger numbers of strains did not support this subdivision, instead identifying a basal group of strains corresponding to clade A2 and a rapidly evolving clone corresponding to clade A1 (3). By contrast, clade B is mainly associated with community-acquired infections (6). Some authors have even recently proposed that it should be reassigned to species E. lactis (7)."

Line 73-75: not clear. rewrite the sentence starting "today".

We have modified the sentence as follows (lines 82-86):

"E. faecium bacteremia increased between 2001 and 2014 in Europe and between 2001 and 2010 in the US (10). This increase may have led to an increase in the use of broad-spectrum antibiotics (e.g., vancomycin) as an empirical option in hospitals, in the absence of rapid results concerning antibiotic susceptibility."

Line 76: inadequate treatment does not make any sense. Did you imply treatment failure?

We have changed the sentence (see reviewer 1's comment above).

Line 80: required sounds mandatory. Replace with "helpful". Rewrite the sentence starting with "In addition" as it does not pass the idea well.

We modified the sentence as follows (lines 92-95):

“The clinical and epidemiological data associated with resistance (e.g., nosocomial acquisition) (14) must be taken into account when selecting the most appropriate treatment. However, tools accelerating the identification of enterococcal species could also provide a robust and complementary means of improving this choice.”

Line 105-106: When you do your second part of the analysis, does this mean you used multiple episodes from same patient? Its not acceptable to get multiple isolate per patient within a calendar year for the sensitivity criteria (Read antibiogram construction in CLSI) . In addition, single patient multiple samples can have variable AST over the time. Explain how you overcame this issue.

The second part of the analysis corresponds to the paragraph results entitled “Characteristics of EFM-S and EFM-R isolates”. In this part, our goal was to compare EFM-S and EFM-R isolates in terms of sources. In this context it was essential to consider all the isolates from the same patient. Indeed, two isolates may be concomitantly obtained from the urine and from the blood of the same patient. Keeping only one of them would arbitrarily change the proportion of sources in the population of isolates to be compared.

Regarding the question of variable ASTs for the same patient, we have only encountered this issue in one patient during two different episodes. In this latter case, only the first episode was considered in the first part of the analysis (i.e. Clinical characteristics of patients infected with EFM-S and EFM-R). In the second part of the analysis both isolates were considered and both were isolated from urine.

We have added more explanation about the differences between these two parts in the method section:

Lines 117-118:

“We first focused on the clinical characteristics of the patients infected with EFM-S or EFM-R isolates. We took into account only the first episode for each patient included”

[...]

Lines 128-133:

“We then compared the sources of isolates and the polymicrobial nature of the infection as a function of ampicillin susceptibility. In this context, we included all isolates from all episodes occurring in all patients over the study period. Only redundant isolates obtained from the same clinical sample during a given episode were excluded. We encountered no cases of differences in susceptibility between isolates collected from the same source during the same episode.”

Line 109: supplemental table .1- you have used only 55 clinical isolates from your study and used previously published strains for the rest. In addition, explain how and when you used 283 patients' data and how many of them are multiple isolates from a single patient?

We thank the reviewer for his careful rereading. Indeed, there was a typo in supplementary table 1. This table was supposed to contain a detail of the information obtained from the review of patients' electronic medical records. We have therefore amended it.

We now modified the methods section as described above to clearly separate the patient clinical analysis (n=283) from the isolates sources comparison (n=372). The set of 52 isolates was used for WGS and MALDI-TOF MS database validation.

We also added the number of cases with multiple isolates obtained from a single patient in the result section "Characteristics of EFM-S and EFM-R isolates" (lines 234-240):

"These isolates included 151 obtained from a patient from whom another isolate was also obtained. There were 61 patients with multiple isolates. We found that 116 of these multiple isolates were associated with the same episode as the other isolate or isolates from the same patient. In total, multiple isolates were available for 48 episodes: 22 episodes involved isolates from different sources (n=48 isolates), 19 involved isolates from the same source (n=41 isolates), and seven episodes involved isolates from both the same and different sources (n=27 isolates).

Line 150: WGS- specify how many strains you sequenced. It's in the results but not in the methods.

Done.

Line 144-145: brief MSP creation protocols and add as a supplemental material for the benefit of the readership.

We have added a description of the protein extraction required for the MSP creation protocol.

Line 140-153 and 229-243: show common spectra associated with FEM-S and FEM-R. I was unable to open the file in the data base shown in line 233. This could be due to my organization computer restriction. But make sure other can access it. How do you address inability of discriminating isolates between clade A1 and A2 as they are different in Amp sensitivity?

The MSP (Mass Spectrum profiles) can be open using Bruker softwares or even using the open source software Openchrom (<http://www.openchrom.net/>). We now also added the individual mass spectra that were used to create the MSP on Zenodo (new version of the repository: <https://doi.org/10.5281/zenodo.7936571>).

We now provide pictures of part of the spectra that may enable to differentiate clade B from clade A1/A2. As can be seen from these pictures, different peaks should be combined to differentiate the strains. For example in the range of 6300-6900 m/Z a peak at 6600 is specific from A1/A2 strains, but may sometimes be absent (EFM-R-03, EFM-R-11 and EFM-

S-9). This could be combine with other peaks as for example the peak at position 8325 m/Z which is nearly always found in A1/A2 (except EFM-S-08).

We also provide a dendrogram as FigS3, constructed from the differences between the MSP of the strains (Euclidean distance, complete linkage). As shown on this figure, clade B isolates are well separated from clade A1/A2. However, the A1 and A2 clades are mixed in the dendrogram, which explains our difficulty in identifying them specifically.

Line 171: What are the criteria used for the P value 0.15? it not a significant number. Could this be a typo?

The p-value 0.15 was not a typo. It is conventionally accepted that variables with a p-value < 0.15 in the univariate analysis should be included in the multivariate analysis. Some authors even suggest including variables with a p-value of less than 0.2 (PMID: 2910056). It is essential to include these variables to limit confounding effects.

Line 204-205: What is the basis of selecting 52 isolates? if you are to show the clade specific unique sensitivity, you need every single isolate per patient in your cohort. There is no randomized selection criteria. Any explanation for this? Running more numbers of isolates and library prep is not that difficult these days.

EFM-S isolate are quite uncommon (usually less than 10%). Thus, we analysed all EFM-S available in our laboratory (n=26). Then, to limit bias, we paired each EFM-S isolate with an EFM-R isolate according to the sex and age of the patient. If we had more isolates available, we would have analyzed them. That is why we discuss the need for larger studies to be able to include more isolates.

Did you see any other phenotypic difference in clad A1 vs A2? Discuss in the discussion section.

We did not assess phenotypic differences between A1 and A2 as it was out of the scope of this study. (see above the response to the question 2 from the reviewer 2)

July 28, 2023

Dr. Guilhem Royer
APHP - Hôpital Henri Mondor - Département de Microbiologie
Bactériologie
51 avenue du Maréchal de Lattre de Tassigny
Créteil 94010
France

Re: Spectrum04545-22R1 (Ampicillin-susceptible *Enterococcus faecium* infections: clinical features, causal clades and contribution of MALDI-TOF to early detection)

Dear Dr. Guilhem Royer:

Your manuscript has been accepted, and I am forwarding it to the ASM Journals Department for publication. You will be notified when your proofs are ready to be viewed.

Sincerely,

Vittal Prakash Ponraj Ph.D., SM(ASCP)
Editor, Microbiology Spectrum
